# Biochemical and Hematological Markers in Workers with Chronical Exposure to Lead and Cadmium in Colombia

**DOI:** 10.3390/toxics10090524

**Published:** 2022-09-01

**Authors:** Fredy Vergara-Murillo, Katiana Martinez-Yanez, Alvaro Fortich-Revollo, Angel Paternina-Caicedo, Boris Johnson-Restrepo

**Affiliations:** 1Grupo de Química y Medio Ambiente, Facultad de Ciencias Exactas, Universidad de Cartagena, Cartagena 130014, Colombia; 2Facultad de Medicina, Universidad del Sinú, Cartagena 130011, Colombia

**Keywords:** occupation exposure, toxic metals, Colombia

## Abstract

The aim of the present study was to find if workers chronically exposed to lead (Pb) and cadmium (Cd) presented changes in their general health and in the clinical parameters of the population under study. We carried out a cross-sectional survey in a sample of informal workers in Cartagena, Colombia. The population under study was composed of male informal workers (≥18 years of age), with experience in their job, selected from occupational settings with potential exposure to Pb and Cd (i.e., mechanics, battery and garbage recyclers, and welders). The median age was 45 years (interquartile range (IQR), 33–53). The median blood Pb level (BLL) was 2 μg/dL (IQR, 0.76–6.22), and the median of blood Cd level (BCL) was 1.22 μg/L (IQR, 0.33–2.01). The study found that 33% of high exposure jobs with BLL > 5 μg/dL *(n* = 57), whereas in ‘control’ workers, this was 15.3% (*n* = 9). The highest BLLs were found in battery recyclers (82.1%; *n* = 23), followed by mechanics (37.3%, *n* = 22). In the logistic regression model adjusted by age, time on the job, smoking and elevated BCL and BLL increased 3.2 times (95% CI, 1.1–9.7) in mechanics and 29.6 times (95% CI, 7.2–145.6) in battery recyclers. This study found negative changes in the health of workers with higher chronic exposure to lead in Cartagena, Colombia.

## 1. Introduction

Chronic exposure to heavy metals is an important source of diseases worldwide [1]. Lead (Pb) and cadmium (Cd) are among the most frequently reported with occupational exposure and are consequently associated to various chronic diseases [2,3,4,5]. Pb and Cd are very important and have a wide range of uses, which help raise standards of living and improve the quality of life for people across the world in the current growing economy. Pb and Cd are used in batteries, airplanes, boats and homes, and are also found in many colorants, paints and metal alloys of protective materials; other uses include printing letterpress and sinkers for fishing net [6]. Occupational exposure to Pb or Cd can occur primarily through inhalation of fumes at work, and accidental ingestion during drinking or eating on site via hands-to-mouth, contact with contaminated hands, as well as wearing dirty clothes [7,8,9] or incorporation [10,11,12], absorption by foreign bodies with Pb retained or embedded (shrapnel, bullets or other foreign bodies with Pb), and to a lesser extent, through the skin [13,14].

Evidence points out that exposure to Pb causes changes in DNA methylation, causing alterations in DNA methyltransferases, increased concentration of oxidative stress, alterations in the endocrine system, metabolic syndrome, increased blood pressure, increased risk of cancer, neurotoxic effects, cognitive dysfunction in child and adults [3,5,15,16,17], renal dysfunction [16], cardiovascular disease [18], hypertension [19], and reproductive organ dysfunction in males and females [20]. Due to the well-known toxicity of Pb in public health, a BLL of concern or blood lead reference value (BLRV) has been lowered from 60 to 3.5 µg/dL between 1960 and 2021. BLRV was 60 µg/dL in the 1960s; between 1970 and 1985 it was <30 µg/dL for children and adults, after evidence of the damage caused by Pb at low concentrations. Between 1985 and 1991, a level of <25 µg/dL was considered safe. After 1991, BLRV suffered a drastic reduction to 10 µg/dL, and by 2012, the Centers for Disease Control and Prevention (CDC) recommended a BLRV of <5 µg/dL for children and adults. Recently, the CDC established a BLRV value <3.5 µg/dL to insure that health injuries do not occur in children. The above considerations had changed because BLRV previously considered normal might cause neurological damage [13].

Cd may inhibit protein synthesis in the human body, alternate carbohydrate metabolism and production of liver enzymes [1]. A chronical exposure to Cd has been reported in the majority of countries [1,21,22,23]. Environmental exposure to Cd occurs via the consumption of food, occupational industries, or smoking [3]. Cd is not an essential element for humans and Cd exposure has a strong impact on human health, affecting liver, kidneys, and bones. Oxidative stress and tissue damage are mechanisms of Cd toxicity [11]. Poisoning with Cd has been linked to several types of cancer, including breast, lung, prostate, nasopharynx, pancreas, and kidney. The organs most sensitive to Cd are the liver and the kidney, in which Cd binds to metallothionein, promoting oxidative stress and generating cellular damage. A target of Cd is the mitochondrion, where it contributes to the accumulation of reactive oxygen species. Cd has shown activity in the hypothalamic pituitary gonadal axis, generating endocrine disruption, and affecting its proper functioning [24]. Cd poisoning can cause neurodegenerative problems relating to reduced senses of smell and hearing, as well as memory [25]. Exposure to Cd has also been suggested as a promoter of polyneuropathy at an older age [6]. One the most important effects of exposure to Cd is in the bones. In rats, osteoporosis has been associated with high levels of exposure to Cd [26], which has also been associated with hypertension in humans and several animal models. However, the mechanisms by which Cd induces hypertension have yet to be completely elucidated. Several potential mechanisms include increased oxidative stress, the disruption of calcium signaling, kidney damage, interference with the renin-angiotensin system, and dysfunction and impairment of the vascular endothelium [12]. Concentrations of Pb and Cd in whole blood is the most common biomarker to indicate human exposure, and certificate reference materials (CRMs) are available to be used for method validation and quality control purposes during the analyses [6].

Around half of the working population in Latin American countries work in informal jobs, which causes concern because these type of workers are associated with high health risks, even at low doses of chronic exposure to Cd and Pb [3,7,27]. Despite this, only a few reports address this problem in Latin America and the Caribbean. In low- and middle-income countries, a large percentage of employment is “informal”; these jobs do not declare income to the state, therefore, they do not have the legal protections and guaranties of government. Cartagena is a tourist and industrial city of over one million people, and as with many other cities worldwide in developing countries, more than fifty percent of people work in informal jobs [21,28]. This study approached informal workers from the point of view of occupations that do not comply with labor laws, since they lack an employee-employer relationship and, therefore, are not covered by the social protection regulations of workers. In many cases, these informal employees access health through the subsidized healthcare regime, and lack a pension or formal savings for their retirement [8]. Informal workers are not prevented from following any occupational health recommendations, but nor are they involved in regular occupational health and safety education programs. These informal workers are more exposed to toxic solids, liquids, and gases, with high concentrations of Cd and Pb [21,27]. In Colombia there is no formal monitoring of heavy metals in the general population, so the effects of chronical exposure to Pb and Cd are unknown.

The associations between Pb and Cd exposure in informal jobs in Latin America and the Caribbean are still important to guide the decision making of health authorities. Therefore, the aim of the present study was to show an association between a workforce chronically exposed to Pb and Cd, changes in the general health, and clinical laboratory parameters.

## 2. Materials and Methods

We carried out a cross-sectional study to identify informal workers in Cartagena, Colombia, and measure the concentrations of Pb and Cd in their blood. This study was carried out between July 2017 and July 2018. Even though Cartagena is a colonial city with widespread tourism and ports, it also has significant industrial and commercial activities. The study protocol was approved by the research ethics committee of the University of Cartagena, registered under the minutes of the meeting held on 11 June 2014 and DB-FM-CERT-2019. Each participant was signed an informed consent form.

### 2.1. Study Population

The study population was composed of 293 male informal workers (≥18 years old), selected from occupational settings with potential exposure to Pb and Cd (i.e., mechanics, battery recyclers, garbage recyclers, and welders). Workers were enrolled in the study after signing a written informed consent form, prior to blood collection, and after a careful description of the study they signed a voluntary consent to participate in the study. The workers also completed a questionnaire about their family’s sociodemographic status, and the behavior and potential sources of Pb and Cd exposure in their work settings. The questionnaire was administered by trained interviewers.

We enrolled informal workers in visits to neighborhoods and commercial districts where informal trades are concentrated. We located mechanics and battery recyclers of related businesses at random. For welders and recyclers, we relied on unions and non-governmental associations that support informal workers to find businesses where these types of workers were concentrated. Control workers (healthcare workers, guards, students, and office workers from the Universidad de Cartagena), were also conveniently chosen from the population of office workers in the University of Cartagena (Cartagena, Colombia).

We chose informal settings that potentially increase the occupational exposure to Pb and Cd. Mechanics handle substances with a high proportion of toxic metals, especially car batteries [12,28]. Battery and garbage recyclers are in contact with urban waste, but especially in the work of recycling car batteries [10]. Welders are exposed to toxic metal fumes, and in the particular matter produced by working with metals [10]. All these informal workers have in common that protection protocols are not frequently followed, and when applied, personal protective equipment is incorrectly used [29]. All the workers in this study have the potential for chronic exposure to Pb and Cd [30]. Additionally, we included other workers with a low potential occupational exposure to Pb and Cd, which we defined as ‘control workers’ [1]. None of the subjects studied were taking any medication. In Colombia, all neighborhoods from each city are classified according to their socio-economic level between 1 and 6 (1 is very poor, 2 is poor, 3 is middle-class, and 4–6 are upper-middle class). We took this governmental classification and used it to characterize the income of the population [7].

### 2.2. Data Collection

The selected workers were visited at their workplace for an interview and later invited for blood collection. The data collected included age, sex, height, and weight (Table 1). In face-to-face interviews, they were asked whether they smoked, current job information, the time dedicated to their occupation, level of education, and years of experience in their jobs. Venous blood samples were collected by trained personnel, in Vacutainer^®^ tubes (USA), with Ethylenediaminetetraacetic acid (EDTA) to determine Pb and Cd in whole blood [6]. The samples were stored at –20 °C until analyzed. For the measurement of systolic and diastolic blood pressure, we followed the recommendations of the World Health Organization (WHO) [31]. The mean of the two blood pressure measurements was considered the value of the blood pressure [32]. The biochemical analysis included total cholesterol and HDL. Blood glucose levels were determined using a Bio Systems kit^®^. White blood cell counts, red blood cell counts, hematocrit, and mean corpuscular volume using heparinized microtubules and a micro centrifuge were measured following a visual method recommended by the WHO [33]. 

### 2.3. Blood Pressure, Hematological and Biochemical Estimations

A fraction of the blood samples collected into the tubes without anti-coagulant were centrifuged at 3400 rpm for 5 min at 20 °C to obtain serum for biochemical analyses. The biochemical analysis included total cholesterol (TC), high-density lipoproteins (HDL), and blood glucose, which were determined using commercial diagnostic Kit-Bio Systems (Costa Brava, Barcelona-Spain). Hematological studies, including white blood cell (WBC) counts, red blood cell (RBC) counts, hemoglobin (Hb), hematocrit (Hct) and mean corpuscular volume (MCV), were carried out using Wright’s stain (Albor Chemical, Bogota Colombia). For systolic and diastolic blood pressure, we considered two measurements with an interval of two minutes, and the mean of both values was considered the value of the blood pressure, systolic blood pressure or diastolic blood pressure. Metabolic syndrome was estimated in workers who met three or more of the following criteria; waist circumference (>102 cm), triglycerides (>150 mg/dL), systolic (>130 mmHg) or diastolic (>85 mmHg) blood pressure, HDL (<40 mg/dL) and glucose (>100 mg/dL), following the US Adult Treatment Panel III of the National Cholesterol Education Program (ATP III) [34,35]

### 2.4. Chemicals and Reagents

Reagents and solvents used were analytical grade. Nitric acid, triton X, Ammonium dihydrogen phosphate, and magnesium nitrate were purchased from Merck^®^ (Darmstadt, Germany). All solutions were prepared in plastic and glassware and were cleaned by soaking in a nitric acid solution (10% *v*/*v*) for 24 h before analysis, then rinsed with deionized water before use. The stock standard solutions of Pb and Cd (1000 mg/L, Merck^®^, Germany) were prepared for analysis by adding a standard solution of nitric acid (0.2% *v*/*v*). The working standard solutions were prepared by the appropriate dilution of stock solutions. For the determination of Cd and Pb in whole blood, we optimized the method to obtain the best sensibility of the instrument using chemical treatment with magnesium nitrate and ammonium dihydrogen phosphate as matrix modifiers, respectively [35].

### 2.5. Instrumental Analysis

The analytical methods used to determine Pb and Cd in whole blood samples were based upon an Atomic Absorption Spectrometer (AAS) iCE 3500 series equipped with a graphite furnace atomizer, Zeeman-effect-background correction and auto-sampler (Thermo Fisher Scientific, Cambridge, UK), which were described in the Appendix A). In brief, pyrolytically-coated graphite tubes were used for all determinations. Calibration curve standards of 2.5, 5, 10, 20, 40, and 60 µg/dL for Pb, and 1, 2, 3, 4, and 6 µg/L for Cd were used. These standards were prepared with a 0.2% nitric acid solution. Standards and whole blood samples were both diluted (1:10) with an aqueous matrix modifier solution (containing 0.2% nitric acid, 0.5% Triton X-100, with ammonium dihydrogen phosphate (0.8 µg/mL) for the Pb and magnesium nitrate (1 mg/mL) for the Cd). The Pb and Cd standards were prepared from standard solutions of 1000 ppm [36].

### 2.6. Quality Control and Quality Assurance

We determined accuracy, precision, linearity, the method detection limit (MDL), and the method quantification limit (MQL), to validate these methods to determine Pb and Cd concentrations in whole blood. Three concentrations of blood-based standard certificate materials (CRM) provided by the New York State Department of Health (Wadsworth Center, Albany, NY, US) were used to validate the methods [27]. For all samples, analyses were performed in duplicate and CRM were analyzed for quality control purposes. The method detection limit (MDL) and method quantification limit (MQL) were estimated in this study from blanks with values of 0.31 and 1.3 µg/dL for Pb and 0.41 and 1.36 µg/L for Cd, respectively (Appendix A). Results of the three different CRMs analyzed in triplicate were CRM name (mean ± SD, recovery percentage), BE 12-12 (2.32 ±0.22 µg/L, 97%), BE 12-14 (2.8 ±0.29 µg/L 87%), BE 12-15 (12.37 ± 0.64 µg/L, 109%) for Cd and BE 12-11 (9.32 ± 1.0 μg/dL,111%), BE 12-13 (4.56 ± 0.48 μg/dL, 108%) and BE 12-14 (25.7 ± 0.7 μg/dL, 99%) for Pb (Appendix A). Calibration curves were considered optimal if the regression coefficient was ≥0.995. Precision was generally <5% Relative Standard Deviation (RSD). Additional information was described in the Appendix A. 

### 2.7. Statistical Analysis

We described the continuous variables of the sample with the median and ranges or IQR, and categorical variables with percentages. Given the non-normality of data, comparisons of continuous variables across groups were made with the Kruskal-Wallis test, while for categorical variables, the chi-squared test was used. We tested for the assumptions of the Kruskal-Wallis test by running the Bartlett test to detect heteroscedasticity, and visually inspected the median with box-plots to assess distributional changes in the medians.

A parsimonious logistic regression was also fitted to assess the effect of each type of informal job on high BCL and BLL. In the logistic regression, we defined levels above 5 µg/dL of Pb and 5 µg/L of Cd as high. The crude model only included work type as exposure variables, with high levels of Pb and Cd as outcome variables. This regression was adjusted by confounding variables (age, time of experience in the job, smoking, education, and socioeconomic status). A *p*-value *<* 0.05 was considered statistically significant for all analyses. All statistical analyses were performed in *R (v4.1.2).*

## 3. Results

### 3.1. Population Characteristics

A total of 293 workers were studied, all of whom were living in Cartagena, Colombia. The age range was between 18 and 77 years old, while the median age was 45 years (IQR, 33–53); prevalence of smoking was 13.4% (*n* = 37). In the population presented, mechanics, garbage recyclers, and battery recyclers, had the lowest education level in the informal worker group with 52.5, 87.5, and 71.4% at elementary school level, respectively, while welders had the highest education level with 9.92% at elementary school level and 44.6% at technical/university level (Table 1). Also, the control worker group had a higher education level (*p*-value < 0.001) and the higher frequency of middle-class participants (p-value < 0.001) in regard to higher exposure group. Smokers were much less frequent than non-smokers and the values ranged from 2.26% for mechanics to 26.9% for battery recyclers.

### 3.2. Laboratory and Anthropometric Parameters

The population with the highest level of cholesterol and triglycerides were the controls, with medians of 196 mg/dL (IQR, 126–284), and 134 mg/dL (IQR, 60–224), respectively. Platelet cell counts in controls workers were 214 cells/L (IQR, 169–382), while the lowest of these counts were recorded in mechanics, with 191 cells/L (IQR, 152–326), followed by welders, with 197 cells/L (IQR, 153–321). Control workers had 14.6% (*n* = 6) of metabolic syndrome, whereas the subjects with the highest frequency of metabolic syndrome were battery recyclers 38.9% (*n* = 7), followed by garbage recyclers (31.6%, *n* = 12) (*p* = 0.171). The median level of glucose in control workers was 79.0 mg/dL (IQR, 60.6–122), yet the highest level of glucose appeared in welders (median of 89.0 mg/dL; IQR, 61.0–166). The work types that displayed the individuals with highest glycemia frequency (≥ 100 mg/dL) was the welders with 18.2% (*n* = 22), and mechanics with 17% (*n* = 10). The percentage of basophils in control workers was 24.1%, compared to the highest levels among workers in battery recyclers (46.4%, *n* = 13) (*p* = 0.023) (Table 2). The analysis shows that 35% *(n* = 102) of the population have a BMI ˂ 25 kg/m^2^, 45% (*n* = 131) are overweight (25–30 kg/m^2^), and 19.9% (*n* = 58) can be considered obese (≥30 kg/m^2^). The median body mass index in controls was 26.1 (IQR, 16.7–35.0), with no significant differences between type of informal labor (Table 2). The controls had higher cholesterol levels (*p* < 0.001), and lower blood glucose levels (*p*-value < 0.001). The full comparisons are listed in Table 1 and Table 2.

### 3.3. Occupational Exposure to Pb and Cd

The median of the BLL in the entire population was 2 μg/dL (IQR, 0.76–6.22), whereas the median of the whole blood Cd concentrations was 1.22 μg/L (IQR, 0.33–2.01). In the whole population, the frequency of higher BLRV (5 µg/dL) was 29.5% (*n* = 87) and in the control workers it was only 15.3% (*n* = 9). The frequencies of higher than BLRV (5 µg/dL) for BLL of concern for exposure worker groups were 30, 23, 22, and 3%, for welders, battery recyclers, mechanics, and garbage recyclers, respectively (Table 1 and Figure 1). The basophil cell counts were positively and significantly associated and increased with higher BLLs (*p* < 0.001, Figure 2), while platelet counts was negatively associated but significantly decreasing with higher BLLs (*p* = 0.012, Table 3) [13]. The control workers had the lowest median concentrations of Cd in whole blood, but this group of individuals had the greatest frequency for the higher than blood cambium reference value BCRV (5 µg/L) with 10.2% (*n* = 5) in regard to all groups, followed by the mechanics with 8.47% (*n* = 5). The logistic regression analysis adjusted by age, education, time in the job, smoking, and socioeconomic status, showed that BLLs are associated to the type of worker, with OR (95% CI) 29.5 (7.2 to 146) for battery recycler, 3.2 (1.1 to 9.7) for mechanics, 1.4 (0.59 to 3.7) for welders, and 0.55 (0.10 to 2.6) for garbage recyclers. Times increased the exposure to heavy metal with respect to the control group. None of the workers studied significantly increased, with respect to control group, for Cd concentration in whole blood for both crude and adjusted logistic analyses (Table 4).

## 4. Discussion

This study examined the association of BLL with occupational exposure of informal workers, including battery recyclers, welders, mechanics, and garbage recyclers. The median of BLLs in the all population was of 2 µg/dL, where it did not exceed the acceptable BLRV of 5 µg/dL recommended by CDC. However, the median of 15.1 µg/dL and mean ± SD of 15.1 ± 14.0 µg/dL for the group of battery recyclers were the highest values reported among all groups in this study, and the only one that was greater than that of BLRV. Consistent with previous reports of battery workers, the results of the present study were similar to other battery workers from Thailand (15.8 ± 3.2 µg/dL) [37], but lower than workers from Pakistan (69.2 ± 37.7) [38] and from Bangladesh (65.7 ± 26.7) [39]. Battery recycling workers had 29 times more health risk than control workers, whereas Pb concentrations in the blood of mechanics and welders were at a lower than acceptable threshold value of BLRV (5 µg/dL), although they had a higher health risk than the control group. 

A positive and significate association was found between basophil cell counts and BLLs from 234 workers with occupational exposure. Basophils are a type of white blood cell, which is a part of the immune system and is identified visually as a cell with purple-blue spots. A healthy subject only contains up to around 3% or less of basophils among all while blood cells. An abnormally high basophil cell count is called basopenia and it may be a clear sign of Pb exposure. Sudjaroen and Suwannahong conducted a control case study with non-exposure and exposure battery workers, with 33 and 30 subjects, respectively, from Thailand [37]. In spite of basophil cell counts being higher for the exposure group than those for the control group, there was no statistical significance for them. Irawati et al.(2022) examined the presence of basophils in 128 enrolled children aged between 1 and 5 years from Indonesia, finding basophils in 26.6% of the collected samples, although the relationship between basophil cell counts and BLL was not statistically significant [40]. This statistical analysis was adjusted by time in the job to account for the increased risk of adverse health effects of Pb, with more time for occupational exposure. Concentrations similar to those found in our study are potentially toxic [41] and can be harmful to human organs, particularly the liver and kidneys [23,42] and may cause cell damage [42,43]. Population studies showed a decreasing platelet count in individuals exposed to Pb, particularly in acute exposures. The data from this study shows that the presence of basophils is correlated with a higher amount of Pb in workers and controls. The presence of basophils is seen as a screening test to consider environmental or occupational exposure to Pb.

In the present study, a negative yet significant association was found between platelet cell counts and blood Pb concentrations in mechanics, battery recycles, welders, and garbage recycling workers. Barman et al., 2014, reported that platelet cell counts were decreased with higher BLL in 429 workers exposed to Pb during lead-acid battery manufacturing, at a plant located in Tamil Nadu, India [44]. Rahimpoor et al. conducted a case-control study with mining workers from Iran, in which platelet cell counts from the miner exposure group were significantly lower than those in non-exposure workers. However, the results were contrary to those for adults; the platelet cell counts in children from electronic waste recycling areas with high BLL were higher than those compared to the control group [45]. Alterations to the normal range of platelet cell counts (150–400 × 10^9^/L) in the subjects may Pb to thrombocytopenia when levels are below 150 × 10^9^/L, and to thrombosis when the count is greater than 400 × 10^9^/L. The present study also shows a reduction of 8% in platelet levels of subjects with values higher than 5 µg/dL (BLRV) (Table 3).

Our analyses are adjusted by the education level in the multivariable models for Pb and Cd (Table 4). One of the factors related to the chronic exposure to Pb can be the lack of education, improper handling of metals, and the poor use of occupational protective equipment. In the case of our study, mechanical welders and particularly battery repairer people have a greater exposure to Pb, compared to control workers without a high exposure to Pb, so we infer that the origin of the exposure is occupational; a poor use of personal protection elements are causing an increase in the systemic BLL. Controls with higher BLLs correspond to participants who declared having bullet impacts in their body. The values of Pb in these individuals were 56.9, 50.0 and 31.9 µg/dL.

An important non-occupational source of exposure to Cd is smoking and diet. The results of this study show that despite some workers having high BCL, there is not a relationship between exposure in the types of jobs and Cd concentrations. However, high Cd concentrations in the blood of workers in the control group suggest that exposure to Cd may be linked to cigarette smoking or diet. A person who smokes 20 cigarettes a day is exposed to 1 µg of Cd/day, most of which is absorbed into the lungs, and seafood consumption, such as crabs, oysters, and scallops, widely found in Caribbean coasts, contain relatively high levels of Cd, on average, 5 μg Cd/g wet weight [46,47,48]. Despite not presenting high BCL in the workers studied, it is important to continue biomonitoring these informal workers. In Colombia, there is a broad population working in informal conditions. 

In Colombia, more than half the workers are covered by government social security. Informal workers in Colombia are mostly affiliated to the subsidized health system, therefore, exposure to heavy metals become an economic burden to the health system since these workers are at a higher risk of suffering high-cost pathologies associated with occupational exposure to Cd and Pb. This study presents evidence of the increased risk of adverse health effects of unprotected work in a population in a middle-income country. Forty-six percent of the Colombian population works in informal jobs [28], such as mechanical workers, metalworking, garbage recycling, and battery repairs, among many other trades. The exposure to heavy metals from these activities and trades, along with the lack of personal protective equipment and the indiscriminate handling of materials containing Pb and Cd, is a factor of concern for public health in Colombia and Latin America. The authors hope that this study will Pb to more extensive studies worldwide. 

The lack of high-quality healthcare coverage and low education levels leads to a very scarce use of personal protection elements, and an increase in the potential risk of exposure to contaminants such as toxic metals, making the effects of occupational exposure an important factor in the burden of disease and the years lost due to disability [20]. Pb and Cd are still highly important elements for the development of the economy. Developing countries require clear policies for the management of solid waste resulting from unused equipment, which is increasing every day. Further studies with greater sample sizes and a better representation of the population of workers might provide a more accurate insight on the harmful health effects of Pb, Cd, and other toxic elements in informal workers in Colombia, and worldwide. In addition, our cross-sectional study does not consider the temporality of the association, therefore our analyses are adjusted by the time working on these activities.

## 5. Conclusions

Informal workers dedicated to recycling batteries had higher concentrations of BLL than the all other groups in the present study, and BLRV recommended by the CDC. According to the logistic regression model, battery recycling workers were at a health risk 29 times higher than the control group, followed by mechanics and welders, in a city with one million citizens, out of which approximately 50% of the working population has informal jobs. These informal workers are not covered by the government’s social security, but are mostly affiliated to the subsidized health system. This study shows a significant problem, in which exposure to heavy metals become an economic burden to the health system, since these workers are at a higher risk of suffering high-cost pathologies associated with occupational exposure to Cd and Pb. The problem is documented by this study, carried out in Colombia, but this problem likely occurs in other countries, too. We recommend, based on the results of the study, that legislation and public health actions be oriented towards improving the conditions in which informal workers are involved. Future studies should be held to measure the impact on public health, and the years of disability generated by occupational exposure to these heavy metals. 

## Figures and Tables

**Figure 1 toxics-10-00524-f001:**
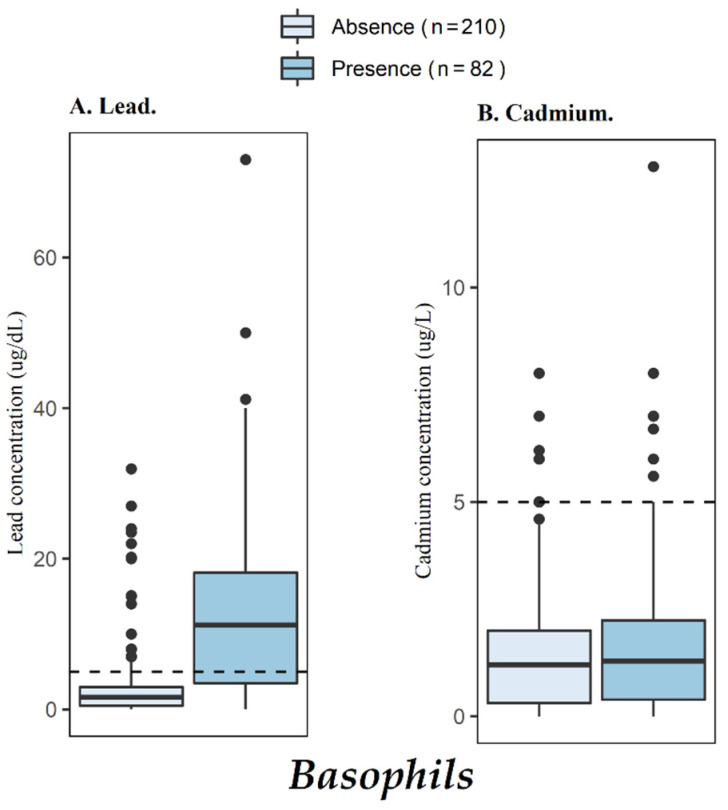
Association between basophil blood cell counts and Pb and Cd concentrations in whole blood. The dotted line indicates the recommended safe limits for Pb and Cd concentrations in whole blood [13].

**Figure 2 toxics-10-00524-f002:**
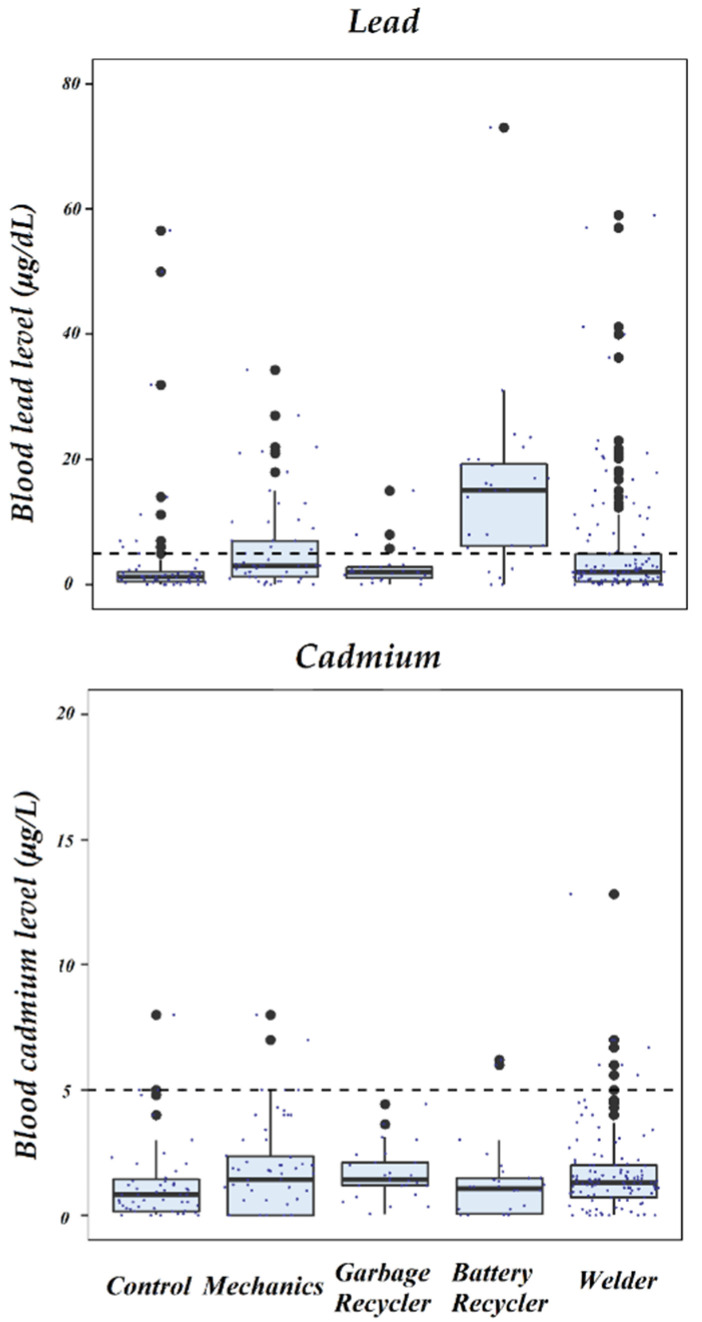
Boxes indicate BLL and BCL by job type, indicating variability outside the upper and lower quartiles. The dotted line refers to the recommended safety limits for Pb and Cd in whole blood [13].

**Table 1 toxics-10-00524-t001:** Summary of data obtained from subjects from Cartagena, Colombia, stratified by type of work.

Parameter	Control (*n* *=* 59)	Mechanic (*n* = 59)	Garbage Recycler (*n* = 26)	Battery Recycler (*n* = 28)	Welder (*n* = 121)	*p*-Value
Age, years, median(range)	43.0 (20.0–77.0)	40.0 (20.0–68.0)	48.0 (21.0–63.0)	39.5 (18.0–63.0)	47.0 (20.0–68.0)	0.006 *
Experience, years, *median(range)*	11.0 (1.00–63.0)	12.0 (0.58–59.0)	20.0 (3.00–50.0)	7.50 (1.00–32.0)	16.0 (2.00–49.0)	<0.001 *
Economic status, %						<0.001 *
Very poor	64.4	64.4	100	64.3	59.5	
Poor	18.6	30.5	0	28.6	33.1	
Middle-class	15.3	5.1	0	7.1	6.6	
Upper middle-class	1.7	0 (0)	0	0	1 (0.8)	
Education, %						<0.001 *
Elementary	6.8	52.5	88.5	71.4	9.92	
High School	22	33.9	11.5	17.9	45.5	
Technical	69.5	13.6	0	10.7	44.6	
Smoking, %	10.1	5.1	19.2	25	13.2	0.112
Pb, µg/d, median(range)	1.31 (≤MQL–56.6)	3.00 (≤MQL–34.3)	1.96 (≤MQL–15.0)	15.1 (≤MQL–73.0)	2.00 (≤MQL–59.0)	<0.001 *
High Pb, %						<0.001 *
No	84.7	62.7	88.5	17.9	75.2	
Yes (>5 µg/dL)	15.3	37.3	11.5	82.1	24.8	
Cd, µg/L, median (range)	0.82 (≤MQL–8.00)	1.43 (≤MQL–8.00)	1.44 (≤MQL–4.44)	1.06 (≤MQL–6.20)	1.32 (≤MQL–12.80)	<0.001 *
High Cd, %						0.287
No	89.8	91.5	100	92.9	93.4	
Yes (>5 µg/L)	10.2	8.5	0	7.1	6.6	

* (*p* < 0.01), Kruskal-Wallis, significant differences between medians across occupations. BLLs ≤ 5 µg/dL, and BCL ≥ 5 µg/L are safe limits for Pb and Cd concentrations in whole blood. The method quantification limit (MQL) [13].

**Table 2 toxics-10-00524-t002:** Laboratory and anthropometric results in subjects from Cartagena, Colombia, stratified by type of work.

	Normal Values	Control	Mechanic	Garbage Recycler	Battery Recycler	Welder	
**Parameter**	Median (Range)						***p*-Value**
		*n* = 59	*n* = 59	*n* = 26	*n* = 28	*n* = 121	
HDL, mg/dL	40.0–60.0	74.0 (58.0–86.3)	75.3 (50.0–105.0)	N/A	74.8 (47.0–98.6)	75.9 (52.0–91.0)	0.631
Cholesterol, mg/dL	<200	196 (126–284)	194 (94.0–298)	156 (93.2–251)	186 (100–248)	181 (112–299)	<0.001 **
Triglycerides, mg/dL	<150	134 (60.4–224)	144 (51.0–348)	112 (44.4–245)	141 (75.1–249)	141 (59.0–255)	0.030 *
Glucose, mg/dL	100.0	79.0 (60.6–122)	84.6 (69.5–139)	87.3 (56.6–165)	81.5 (68.0–132)	89.0 (61.0–166)	<0.001 **
BMI	20.0	26.1 (16.7–35.0)	25.8 (17.9–36.9)	25.2 (22.6–29.3)	26.4 (17.0–39.5)	26.9 (19.0–36.4)	0.087
SBP, (mm Hg)	<120	130 (90–190)	122 (90–170)	120 (100–160)	127 (110–190)	118 (95–177)	<0.001 **
DBP, (mm Hg)	<130	83 (60–110)	78 (60–102)	81 (60–100)	80 (65–143)	78.0 (55–105)	0.010 **
Abd Circ, cm	<95.0	103 (75.0–135)	97.0 (73.0–136)	110 (92.0–130)	103 (90.0–133)	99.0 (77.0–136)	<0.001 **
Hip-waist index	0.96–0.99	0.92 (0.80–1.16)	0.92 (0.80–1.14)	0.91 (0.88–1.03)	0.90 (0.86–1.10)	0.92 (0.82–1.04)	0.427
Platelets/10^9^, cells/L	150–450	214 (169–382)	191 (152–326)	201 (170–326)	236 (159–391)	197 (153–321)	0.009 **
Hemoglobin, mg/dl	13.8–17.2	14.6 (11.6–17.3)	15.0 (13.0–16.3)	14.6 (11.6–16.3)	14.4 (12.0–16.3)	15.0 (10.6–17.3)	0.038 *
Hematocrit,(vol% of RBC)	40.0–54.0	44.0 (35.0–52.0)	45.0 (32.0–49.0)	44.0 (35.0–49.0)	44.0 (38.0–50.0)	45.0 (32.0–52.0)	0.047 *
Basophils, %	0–3	24.1	39.0	17.9	46.4	24.0	0.023 *
MetS, %	N/A	14.6	29.6	31.6	38.9	22.1	0.171

Significant differences * (*p* < 0.05) and ** (*p* < 0.01). HDL: High-density lipoproteins, MetS: Metabolic syndrome. Abd. Circ: Abdominal circumference. BMI: Body mass index. SBP: Systolic blood pressure. DBP: Diastolic blood pressure. N/A: Not available. RBC: Red blood cells.

**Table 3 toxics-10-00524-t003:** Key parameters for the sample of informal workers, stratified by elevated (≥5 µg/dL) BLL.

Key Parameters	Non-Missing Data	All Patients	Normal Pb Levels	Elevated Pb Levels	Effect of Abnormal Lead	*p*-Value
	** *n* **		**Percentege *(%)***		**OR (95% CI)**	
Basophile	294	28.6	11.6	69.0	12.3 (6.62 to 23.7)	<0.001
Smoking	276	13.4	14.6	10.7	0.81 (0.34 to 1.78)	0.499
			**Median (range)**		**β (95% CI)**	
BMI	291	26.4 (15.4–39.5)	26.3 (15.4–36.9)	26.8 (17.5–39.5)	0.19 (−0.91 to 1.30)	0.314
HDL, mg/dL	219	75.7 (47.0–105.0)	75.7 (50.0–105.0)	76.0 (47.0–98.6)	1.61 (−8.42 to 11.64)	0.857 *
Cholesterol, mg/dL	294	185 (93–299)	187 (93–284)	178 (100–299)	3.86 (−0.80 to 8.52)	0.294
SBP, mm Hg	272	122 (90–190)	120 (90–190)	125 (95–190)	1.93 (−1.06 to 4.92)	0.160
DBP, mm Hg	272	80 (55–143)	80 (60–110)	80 (55–143)	−6.61 (−23.16 to 9.94)	0.710 *
Platelets/10^9^, cells/L	181	201 (152–391)	210 (155–382)	193 (152–391)	−2.55 (−6.74 to 1.63)	**0.012**
Glucose, mg/dL	306	86 (57–166)	86 (61–166)	86 (57–140)	−0.41 (−1.45 to 0.62)	0.367
Hematocrit, (vol% of RBC)	188	44.0 (32.0–52.0)	44.0 (32.0–52.0)	44.0 (32.0–50.0)	0.19 (−0.91 to 1.30)	0.884

BMI: Body mass index. SBP: Systolic blood pressure. DBP: Diastolic blood pressure. HDL: High-density lipoprotein significant differences (*p* < 0.05).

**Table 4 toxics-10-00524-t004:** Adjusted association between job type and high BCL and BLL, using logistic regression.

Job Types	Cd		Pb	
	Crude OR (95% CI)	Adjusted OR (95% CI)	Crude OR (95% CI)	Adjusted OR (95% CI)
Control workers	Reference	Reference	Reference	Reference
Mechanic	0.82 (0.22 to 2.87)	0.71 (0.12 to 4.83)	3.30 (1.40 to 8.34)	3.20 (1.13 to 9.74)
Garbage recycler	N.E.	N.E.	0.45 (0.10 to 1.63)	0.55 (0.10 to 2.57)
Battery recycler	0.68 (0.10 to 3.19)	0.75 (0.07 to 7.03)	25.56 (8.29 to 93.85)	29.55 (7.17 to 145.64)
Welder	0.63 (0.21 to 1.98)	0.61 (0.13 to 3.43)	1.83 (0.83 to 4.37)	1.42 (0.59 to 3.73)

Logistic regression analysis was adjusted the using variables of age, time experience in the job, smoking, education, and socioeconomic status. N.E.: Not estimable.

## Data Availability

The data is available upon reasonable request to the corresponding author.

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
