# Peer review of "Biochemical and Hematological Markers in Workers with Chronical Exposure to Lead and Cadmium in Colombia"

_toxics, 2022, doi:10.3390/toxics10090524_

Round 1
Reviewer 1 Report (New Reviewer)
The study evaluates exposure to Pb and Cd in informal workers from Cartagena Colombia. The study is well design, reports key sampling and analytical methods and very relevant for the country.
Some minor revisions suggested:
1. Line 349 add a space between 1985 and to
2. The study was conducted in Colombia, what are the national threshold limits for Pb and Cd in blood.
3. The introduction should report a rationally for selecting blood as a main biomarker of exposure and for the measurements of blood pressure, hematological and biochemical estimations.
4. for the section 3.3 report exposure as concentrations instead of percentage.
5. Figures 1 and 2 describe the boxplots
6. What was the DF for Pb and Cd? how did you treat the non detected?
7. Table 1, summary of data:
Add data for all the participants
What does the n(%) means? I can see the data reported by percentage
8. Can tables 2 and 3 be moved to the SI?
9. For the discussion, the data should be reported as concentrations, high levels makes it difficult to follow
10. Line 321 change patients for workers or participants
11. Add a figure or table comparing concentrations of this study with other sites (for example e-waste workers) can give a better insight of the situation. Also, how the concentrations compare with the general population of Colombia?
12. Add a table of contents to the SI
13. Add reference to the method, is this method validated?
14. Table S1. reports that the number of experiments are in parenthesis, I can't see any of that information on the table. What experiments? How was the MDL and MQL calculated?
15. Table S.3 Why the accuracy was calculated with goat whole blood and not human blood? how relevant is the matrix? did you use any human whole blood to calculate recoveries? how many times did you measure the reference material?
16. Table s4. Why was it necessary to spike the CRM? do you have recoveries for real world samples?
Author Response
- Line 349 add a space between 1985 and to
Response. It was changed according the suggestion.
- The study was conducted in Colombia, what are the national threshold limits for Pb and Cd in blood.
Response. There is not national threshold limit concentrations for Pb and Cd in whole blood. Therefore, where following the US-EPA reference values (RfD) which was mention in the manuscript
- The introduction should report a rationally for selecting blood as a main biomarker of exposure and for the measurements of blood pressure, hematological and biochemical estimations.
Response. We agree with the reviewer. So, we included a sentence in the introduction showing why whole blood is the most common matrix to determine the concentrations of Pb and Cd. Therefore, heavy metal concentrations such as Pb and Cd in whole blood are used excellent biomarker of exposure. Also, method validations were developed using reference materials before starting with analyzes
- for the section 3.3 report exposure as concentrations instead of percentage.
Response. To provide greater clarity, it was considered the blood lead reference value (BLRV) for higher concentration than 5 µg/dL and blood cadmium reference value (BCRV) for higher concentrations than 5 µg/L.
- Figures 1 and 2 describe the boxplots
Response. It was rewrite the sentence as follow: “In the boxplot, the basophil cell counts was positively and significantly associated in-creased with higher Pb concentrations (P<0,001, Fig. 1) while platelet counts was negatively associated but significantly decreasing with higher Pb concentrations (p=0.012, Table 3)”
- What was the DF for Pb and Cd? how did you treat the non-detected?
Response. The non-detected samples were replaced in the Table 1 as values below of Method Detection Limit (MDL) for Pb and Cd.
- Table 1, summary of data: Add data for all the participants
What does the n(%) means? I can see the data reported by percentage
Response. The Table 1 was reorganized for a better reading
- Can tables 2 and 3 be moved to the SI?
Response. There is no many tables in the manuscript. Therefore, Tables 2 and 3 are kept because they are important to explain the relation between the informal jobs, metals and health variables in the subjects into the study
- For the discussion, the data should be reported as concentrations, high levels make it difficult to follow
Response: We rewrite the text to improve the discussion and more clarify the high levels of Cd and Pb, It was used the percentages to know the impact on workers who exceed the recommended levels of metals
- Line 321 change patients for workers or participants
Response. The word “workers” was included in the entire manuscript.
- Add a figure or table comparing concentrations of this study with other sites (for example e-waste workers) can give a better insight of the situation. Also, how the concentrations compare with the general population of Colombia?
Response: The purpose of the study is to reveal the levels of heavy metals in the blood as an indicator of exposure to heavy metals in these informal jobs. The levels found are contrasted with the studies and recommendations of academic and health authorities.
- Add a table of contents to the SI
Response: We welcome the suggestion, clarify the terms, and reorganize the information in the supplemental material.
- Add reference to the method, is this method validated?
Response: The method was validated and the validation information was included in the section 2.6 of Quality control and quality assurance. Additional information was place in the supplementary information.
- Table S1. reports that the number of experiments are in parenthesis, I can't see any of that information on the table. What experiments? How was the MDL and MQL calculated?
Response: MDLs (method detection limit) and the method quantification limit (MQL) were calculated as 3 and 10 times the standard deviation of the reagent blanks, respectively. MDL and MQL were estimated analyzed a set of six blanks.
- Table S.3 Why the accuracy was calculated with goat whole blood and not human blood? how relevant is the matrix? did you use any human whole blood to calculate recoveries? how many times did you measure the reference material?
Response: Goat blood is the most appreciated standard reference material for human blood, it has a wide development that includes the determination of metals in more than 10 laboratories around the world by different techniques, in the validation of the method we used, human blood for the elaboration of the curve by the added standard method and the CRM were used to verify the response of the curve at low, medium and high concentrations. The reference material was measured before each batch and during the reading every 10 samples to verify the quality of the readings.
- Table s4. Why was it necessary to spike the CRM? do you have recoveries for real world samples?
CRMs contain the concentrations the heavy metals of our interest and were analyzed validation the method and quality control purpose. It was added the following paragraph as follows:
“All samples are analyzed in duplicate and the value of the area under the curve is used to determine the concentrations of the metals. Before each analysis, the calibration curve was made with human blood by the added standard method and for quality control measurements of three CRM were made in each run. a reagent blank and an air sample were also read each day. if an R2 greater than 0.995 was met and the SRM corresponded to the reference values, whole blood samples were read; every 10 samples, an SRM was included to verify quality during analysis. the samples high concentrations of Cd and Pb, a second reading was made to verify.”
Reviewer 2 Report (New Reviewer)
This paper summarizes the effects of Pb and Cd exposure in "informal" workers in Columbia. The authors were able to collect blood samples from workers with different at risk occupations and demonstrate that at least for Pb exposures are elevated relative to the general population. The methodology is apparently acceptable. Given the lack of information on exposure metrics for at risk populations in developing countries this work is important and merits publication.
It is interesting that that the highest Cd levels were found in the control workers, I'm not sure what this means or if it just reflects a general background Cd level.
Primary issue is with the results and discussion sections. Several one or two sentence paragraphs, this needs to be fixed. Also need to have some better transition sentences between topics in the introduction.
I would be interested to see how the rates of exposure documented here compare to rates of exposure for "formal" workers with the same occupations elsewhere in the world. Perhaps that information does not exist though, hence the need for this study.
Specific line comments:
41: it says 9.3 million years of disability workers, was this supposed to say 9.3 million individuals?
44: A lot of disorders listed, probably needs more references.
54: CDC revised blood Pb limit to 3.5 ug/dL in 2021
70: No reference for memory effects from Cd exposure
109: Should provide a protocol number for the ethic committee approval.
148: Wasn't clear if the vacutainers were pre-cleaned. Were blank levels tested?
157: Describe the "manual method"
168: I think a typo here.
305: Typo here, probably "whit" should be "with"
313: Unclear how the study demonstrates kidney, liver, and other cell damage. Maybe I'm just not understanding.
317: Do the basophils indicate cell damage for specific organs? How easy/cheap is the basophil test compared to a blood Pb or Cd test?
336: Isotopic analysis could help evaluate this assumption
340: Those are very high blood Pb levels!
343: Describing more results without any discussion of significance. This is basically a repeat of the results section here.
348: Was there a significant difference for smokers? I don't find that in the paper.
Author Response
Comments and Suggestions for Authors
This paper summarizes the effects of Pb and Cd exposure in "informal" workers in Columbia. The authors were able to collect blood samples from workers with different at risk occupations and demonstrate that at least for Pb exposures are elevated relative to the general population. The methodology is apparently acceptable. Given the lack of information on exposure metrics for at risk populations in developing countries this work is important and merits publication.
It is interesting that that the highest Cd levels were found in the control workers, I'm not sure what this means or if it just reflects a general background Cd level.
Primary issue is with the results and discussion sections. Several one or two sentence paragraphs, this needs to be fixed. Also need to have some better transition sentences between topics in the introduction.
I would be interested to see how the rates of exposure documented here compare to rates of exposure for "formal" workers with the same occupations elsewhere in the world. Perhaps that information does not exist though, hence the need for this study.
Response: We hope that the changes made meet your expectations.
Specific line comments:
41: it says 9.3 million years of disability workers, was this supposed to say 9.3 million individuals?
Response: It has been modified this text as follows
“Pb is a problem for human health, with a mortality burden for 2017 of about half a million people, 9.3 million disability-adjusted life years worldwide and 12,4% of the global burden of idiopathic developmental disability”
Disability-adjusted life years (DALYs) for a disease or health condition are calculated as the sum of the Years of Life Lost due to premature mortality in the population and the Years Lost due to Disability for people living with the health condition or its consequences. One DALY can be thought of as one lost year of “healthy” life.
44: A lot of disorders listed, probably needs more references.
Response: It was added a couple more references to support the above
54: CDC revised blood Pb limit to 3.5 ug/dL in 2021
Response: it was considered at the time of the review, we have taken into account 5 µg/dL because it is the limit allowed in adults, the values ​​of 3.5 µg/dL are considered for children
70: No reference for memory effects from Cd exposure
Response: It was added the reference for this line.
109: Should provide a protocol number for the ethic committee approval.
Response: It has been modified this text as follows
“The study protocol was approved by the research ethics committee of the University of Cartagena, registered under the minutes of the meeting held on 11th June 2014 and DB-FM-CERT-2019. Before sampling, each participant was informed of the potential risk and benefits of this research and signed an informed consent form”.
148: Wasn't clear if the vacutainers were pre-cleaned. Were blank levels tested?
Response: It was added this text in the supplementary Information, as follows
“All samples are analyzed in duplicate and the value of the area under the curve is used to determine the concentrations of the metals. Before each analysis, the calibration curve was made with human blood by the added standard method and for quality control measurements of three SRM were made in each run. a reagent blank (all reactives mixed in a vacutainer with EDTA) and an air sample were also read each day. if an R2 greater than 0.995 was met and the SRM corresponded to the reference values, whole blood samples were read; every 10 samples, an SRM was included to verify quality during analysis. the samples high concentrations of Cd and Pb, a second reading was made to verify.”
157: Describe the "manual method"
Response: It was added some sentence on the text, as follows in the methodology
“Hematological studies, including white blood cell (WBC) counts, red blood cell (RBC) counts, hemoglobin (Hb), hematocrit (Hct) and mean corpuscular volume (MCV) were carried out using Wright´s stain”.
168: I think a typo here.
Response: we agree.
305: Typo here, probably "whit" should be "with"
Response: we agree.
313: Unclear how the study demonstrates kidney, liver, and other cell damage. Maybe I'm just not understanding.
Response: It was added some sentence on the text, as follows
“Similar concentrations to those found in our study are potentially toxic [37] and can be harmful to human organs, particularly the liver and kidneys [21,38], and be capable of cause cell damage [38,39]”.
317: Do the basophils indicate cell damage for specific organs? How easy/cheap is the basophil test compared to a blood Pb or Cd test?
Basophils cell counts can be performed manually by an experimented biomedical professional or using a machine, allows to see the basophilic dotted, it is a blood smear with wright staining accessible to laboratories that only have a microscope, the analysis of Pb by atomic absorption is much more expensive and complex
336: Isotopic analysis could help evaluate this assumption
Response. We strongly agree that an isotopic analysis can determine the origin of the elements to which informal workers are exposed, in the case of lead we found significant differences between the controls and the informal jobs evaluated, given this fact and the scarce use of personal protection materials, we assume that the exposure is occupational, it will derive from their work activities.
340: Those are very high blood Pb levels!
Response: We agree. It caused us concern and for this reason the participants were verified and contacted to find out what could be the cause of these levels of Pb, these people had bullets in their bodies, which could explain such high levels of Pb
343: Describing more results without any discussion of significance. This is basically a repeat of the results section here.
Response: We are very grateful for your suggestion and have made the respective changes in the discussion
348: Was there a significant difference for smokers? I don't find that in the paper.
Response: We are very grateful for your suggestion and have made the respective changes in the discussion
Response. It was added the sentence in the section 3.1 of population characteristics as follows:
“Compared to smokers, the smokers were much lower than the non-smokers and the values ranged from 2.26% for mechanics to 26.9% for battery recyclers.”
Reviewer 3 Report (New Reviewer)
Revision of the manuscript “Changes in metabolic and hematological parameters in workers 2
with chronic occupational exposure to lead and cadmium in Colombia”
Lead (Pb) and cadmium (Cd) have raised highh concerns as potential hazards for human health escpecially within long term toxic effects. However, there are still many profesions and occupational conditions leading to high exposure risks. From this point, this research has notable actuality and importance.
Within discussion and introduction, also check OSHA directives,
DIRECTIVE (EU) 2022/431 OF THE EUROPEAN PARLIAMENT AND OF THE COUNCIL of 9 March 2022 amending Directive 2004/37/EC on the protection of workers from the risks related to exposure to carcinogens or mutagens at work
https://eur-lex.europa.eu/legal-content/EN/TXT/PDF/?uri=CELEX:32022L0431
It would be reccomended to reckeck the Introduction part, description of the significance and aim of the study as there are many self repeating sentences used.
The methods should be rechecked.
Line 152-153: please describe at which period the measurements were provided of the blodd pressure
There is much information provided in Tables 1 -3
There should be more information provided on the evaluation of these data.
The information provided in Table 4 (the reggression analysis) should be supported by some graphs, which could be provided in the Supplementary material.
It would be nice to show the R2 values in order to better emphasize the correlation data.
Īt is not clear, whether there was any correlation between the PB and the objective measurements provided by the authors of this study.
The reviewer would like to reccomend
Before submitting, it is reccomended to give the manuscript and also the supple mentary material to native English speaking reviewer. The Supplementarry opart should be checked.
The additional minor corrections are suggested as follows:
1) For the reagents and other mmaterials, please indicate the providers and the country.
2) In the Supplementary material, the section Method for determination of Lead and Cadmium in whole blood should be corrected. It is randommly written.
3) Īn the manuscriipt and also seperately in the supplementary materials, indicate the acronyms used
4) For example, BE 12-12, BE 12-14, BE 12-11, BE 12-13 and BE 12-14 should be desrcibed also in the Supplementary material.
5) Table S1: There are no parenthesis provided as mentioned in the Table caption: Table S1. Average limits of detection and quantification in whole blood analysis by atomic absorption spectrometry. The number of experiments is in parenthesis
6) It is not clear, what tipe of the method was used for the evaluation of method sensitivity (MDL: method detection limit; MQL: method quantification limit).
7) Commonly MQL is aproximatelly 3 x MDL. Here MQL is aproximatelly 2 x MDL
8) In the whole text and the Supplementary material, please check the significant figures.
9) Table S2 – there are concentrations provided, but RSD values are not mentioned (they are provided, but they are different that rise concennrns about the method validation)
10) Table S3: terms ppb are used. Please in the whole text and the Supplementary material the same measure units should be provided.
11) In the abstract, correct the language
12) IQR acronym should be recognised within the first use of it.
13) It is not clear from the manuscript, what was the control
14) Line 23, please descibe more clearly “ significant adverse changes” by the results
15) Line 30: toxic agents
16) Keywords: Toxic metals
17) Lines 32-33: rephrase “due to the benefits Pb brings to the quality of 32
18) life of modern mankind and to the world economy.”
19) Lines 44, 46 “cardiovascular disease” repeated
20) Line 149: vacutainer© tubes should be replaced by Vacutainer® tubes (Producer, Country) ?
21) Fig. 1 and elsewhere in the text: please check - ug is used in the place of µg
22) Fig. 2 - please place the legends out of the graph.
Author Response
Comments and Suggestions for Authors
Revision of the manuscript “Changes in metabolic and hematological parameters in workers 2
with chronic occupational exposure to lead and cadmium in Colombia”
Lead (Pb) and cadmium (Cd) have raised high concerns as potential hazards for human health especially within long term toxic effects. However, there are still many professions and occupational conditions leading to high exposure risks. From this point, this research has notable actuality and importance.
Within discussion and introduction, also check OSHA directives,
DIRECTIVE (EU) 2022/431 OF THE EUROPEAN PARLIAMENT AND OF THE COUNCIL of 9 March 2022 amending Directive 2004/37/EC on the protection of workers from the risks related to exposure to carcinogens or mutagens at work
https://eur-lex.europa.eu/legal-content/EN/TXT/PDF/?uri=CELEX:32022L0431
It would be recommended to recheck the Introduction part, description of the significance and aim of the study as there are many self-repeating sentences used.
Response. The introduction was revised completely and the aim of the study the study was improved.
Line 152-153: please describe at which period the measurements were provided of the blodd pressure
Response: Units of mm Hg for BP were added in the Table 2 and 3
There is much information provided in Tables 1 -3
Response: The Tables 1-3 were checked and removed all the information that it was not necessary
There should be more information provided on the evaluation of these data.
Response: Additional clarification was included in the text for the Figure 1-3
The information provided in Table 4 (the regression analysis) should be supported by some graphs, which could be provided in the Supplementary material.
Response. Unfortunately, any additional graphs are not provided for the statically packaged in R for the logistical regression
It would be nice to show the R2 values in order to better emphasize the correlation data.
Response: We are grateful for your suggestion, we can tell you that in this study, it was not used Pearson or Spearman regressions. It was used the logistic regression model to evaluate whether or not an event occurred. So, we calculate the odds ratio which are function of the probability. The odds ratio indicates how much more likely a certain even occurs in a group relative to the other. In our study, it was used the binary predictor BLL < BLRV (5 µ/dL) versus BLL ≥ BLRV (5 µ/dL). Odds ratios that are greater than 1 indicate that the event is more likely to occur as the predictor increases
Īt is not clear, whether there was any correlation between the PB and the objective measurements provided by the authors of this study.
Response: In the Figure 2, SPB and DBP variables were statistical significant (P £ 0.05) when exposure groups were compared with control group. However, in the Figure 3, SPB and DBP variables were not significant when were compared BLL < BLRV (5 µ/dL) versus BLL ≥ BLRV (5 µ/dL).
The reviewer would like to recommend
Before submitting, it is recommended to give the manuscript and also the supple mentary material to native English speaking reviewer. The Supplementary apart should be checked.
Response: We strongly agree with the reviewer. It was included a supplementary material with new information which is complementary and important for the manuscript.
The additional minor corrections are suggested as follows:
- For the reagents and other materials, please indicate the providers and the country.
Response: It was included in the supplementary material.
- In the Supplementary material, the section Method for determination of Lead and Cadmium in whole blood should be corrected. It is randomly written.
Response: The methods were written completely and included in the supplementary material.
- In the manuscript and also separately in the supplementary materials, indicate the acronyms used
Response: All acronyms used in the manuscript were place in the supplementary material.
- For example, BE 12-12, BE 12-14, BE 12-11, BE 12-13 and BE 12-14 should be described also in the Supplementary material.
Response: The Table 2S was included in the supplementary material with full information about Blood CRM used.
5) Table S1: There are no parenthesis provided as mentioned in the Table caption: Table S1. Average limits of detection and quantification in whole blood analysis by atomic absorption spectrometry. The number of experiments is in parenthesis
Response: We appreciate the comment, we made the respective changes in the Supplementary information
1) It is not clear, what tipe of the method was used for the evaluation of method sensitivity (MDL: method detection limit; MQL: method quantification limit).
Response: We appreciate your suggestion; we have made the changes in the Supplementary Material
7) Commonly MQL is approximately 3 x MDL. Here MQL is approximately 2 x MDL
Response: We appreciate your suggestion; we have made the changes in the Supplementary Material
8) In the whole text and the Supplementary material, please check the significant figures.
Response: We appreciate your suggestion; we have made the changes in the Supplementary Material
9) Table S2 – there are concentrations provided, but RSD values are not mentioned (they are provided, but they are different that rise concerns about the method validation)
Response: We appreciate your suggestion; we have added the RDS to the Table S2 table
10) Table S3: terms ppb is used. Please in the whole text and the Supplementary material the same measure units should be provided.
Response: We appreciate your suggestion; we have made the changes in the Supplementary Material
11) In the abstract, correct the language
Response: Thank you very much for the observation, we have made the changes in the abstract
12) IQR acronym should be recognized within the first use of it.
Response: We appreciate the comment, we made the respective changes in the abstract
13) It is not clear from the manuscript, what was the control
Response: We appreciate the comment; we clarify in greater detail in the methodology
14) Line 23, please describe more clearly “significant adverse changes” by the results
Response: We appreciate the comment, for clarity we made the change in the abstract as follows
“This study found negative changes in the health of workers with higher chronic exposure to lead in Cartagena, Colombia.”
15) Line 30: toxic agents
Response: We appreciate the comment, we made the respective changes in the introduction
16) Keywords: Toxic metals
Response: We appreciate the comment, we made the respective changes in the keywods
17) Lines 32-33: rephrase “due to the benefits Pb brings to the quality of 32
Response: We appreciate the comment, for clarity we made the change in the introduction as follows: “Pb and Cd are very important by wide uses because raising standards of living and im-proving quality of life across the world in the current growing economy. Pb and Cd are used in batteries, airplanes, boats, homes.”
18) life of modern mankind and to the world economy.”
Response: We appreciate the comment, we made the respective changes in the introduction
19) Lines 44, 46 “cardiovascular disease” repeated
Response: We appreciate the comment, we made the respective changes in the introduction
20) Line 149: vacutainer© tubes should be replaced by Vacutainer® tubes (Producer, Country)?
Response: We appreciate the comment, we made the respective changes in the methodology
21) Fig. 1 and elsewhere in the text: please check - ug is used in the place of µg
Response: We appreciate the comment, we made the respective changes in the fig 1
22) Fig. 2 - please place the legends out of the graph.
Response: We appreciate the comment, we made the respective changes in the graph
Round 2
Reviewer 3 Report (New Reviewer)
Manuscript was corrected and now can be published in the revised form
This manuscript is a resubmission of an earlier submission. The following is a list of the peer review reports and author responses from that submission.
Round 1
Reviewer 1 Report
The manuscript entitled “Changes in the health and laboratory parameters in workers with chronical exposure to lead and cadmium” aims to assess various health-related laboratory parameters and to determine Cd and Pb in blood to estimate possible effects of occupational exposure on the health of workers. The data obtained in the study are of great importance for decision-making and the establishment of regulations and safety measures related to occupational health in Cartagena, Colombia. However, in my opinion, the manuscript is poorly written and needs significant improvement to be of adequate quality for publication in the journal Toxics. My concerns include: 1) the lack of information on the current state of the art regarding occupational exposure to Pb and Cd and health effects in the introduction; 2) the lack of information in the material and methods part (number of participants, were there women in the control group? methods used to determine laboratory parameters have to be described in more details, statistical part needs to be better explained: what were the dependent variables, what was crude, and what adjusted model? how were levels below LOD/LOQ treated? etc.); 3) the results are not sufficiently explained and seem to be reported (described) randomly and not in a logically coherent way (e.g. age, education, smoking are described for the workers while for the control they comment only on BMI, in section 3.2. commented some of the basic characteristics - which are presented in section 3.1, etc.).
In addition, major concerns represent the statistical analyses in the manuscript (especially the logistic regression models) which do not seem appropriate in order to assess the aim of the study, which is the assessment of the health changes in the Pb and Cd exposed workers. Although the data obtained offer great potential, it seems that the authors failed to appropriately assess the possible association of health-related parameters (cholesterol, blood pressure, determination of metabolic syndrome, glucose levels, etc.) with actual Pb/Cd exposure in occupational settings and in controls, and as such, to assess the possible effect of Pb and Cd on the health of workers. It appears that the study rather focuses on Pb and Cd exposure assessment in separate types of work (as seen based on Table 4 and abstract). Furthermore, the English language of the manuscript needs to be significantly improved. Therefore, at this point, my decision would be to reject the manuscript for publication.
Author Response
The manuscript entitled “Changes in the health and laboratory parameters in workers with chronical exposure to lead and cadmium” aims to assess various health-related laboratory parameters and to determine Cd and Pb in blood to estimate possible effects of occupational exposure on the health of workers. The data obtained in the study are of great importance for decision-making and the establishment of regulations and safety measures related to occupational health in Cartagena, Colombia. However, in my opinion, the manuscript is poorly written and needs significant improvement to be of adequate quality for publication in the journal Toxics.
My concerns include:
- the lack of information on the current state of the art regarding occupational exposure to Pb and Cd and health effects in the introduction.
Reviewer #1 - Response #1: We thank the reviewer for the recommendation. We have updated the introduction following your recommendations.
2) the lack of information in the material and methods part (number of participants, were there women in the control group?
Reviewer0020 #1 - Response #2: We thank the reviewer for the recommendation. We only surveyed women as part of our study. The number of participants is listed in section 3.1 (Population characteristics of Results), as follows:
“A total of 295 workers were studied.”
methods used to determine laboratory parameters have to be described in more details, statistical part needs to be better explained: what were the dependent variables, what was crude, and what adjusted model?
Reviewer #1 - Response #3: We thank the reviewer for the recommendation. We added a description of the crude model, the variables used in the adjusted model were: age, time of experience in the job, smoking, education, and socioeconomic status. This sentence was modified as follows:
The crude model only included work type as exposure variables, with high levels of Pb and Cd as outcome variables. This regression was adjusted by confounding variables (age, time of experience in the job, smoking, education, and socioeconomic status, and high lev-els of cadmium and lead).
how were levels below LOD/LOQ treated? etc.);
Reviewer #1 - Response #4: We thank the reviewer for the recommendation. We added a description of these in the supplementary material.
3) the results are not sufficiently explained and seem to be reported (described) randomly and not in a logically coherent way (e.g. age, education, smoking are described for the workers while for the control they comment only on BMI, in section 3.2. commented some of the basic characteristics - which are presented in section 3.1, etc.).
Reviewer #1 - Response #5: We thank the reviewer for the recommendation. We have made some modifications to results, as follows:
3.1. Population characteristics
A total of 295 workers were studied, all living in Cartagena, Colombia. The range of age was between 18 to 77 years old, while the median age was 45 years (IQR, 33-53). prevalence of smoking was 13.4% (n = 37). Most workers studied up to high school (Table 1). The control jobs had a higher frequency of middle-class participants (P-value < 0.001), and a higher education level (P-value < 0.001).
3.2. Laboratory and antropometric parameters
The population with the highest level of cholesterol and triglycerides were the controls with a median of 196 mg/dL (IQR, 126-284) , and 134 mg/dL (IQR, 60.4-224), respectively.
Platelets levels in controls were of 201 cells/L (IQR, 152-391), while the lowest levels of platelets were recorded in mechanics with 191 cells/L (IQR,168-252), followed by welders, with 197 cells/L (IQR,186-206).
Control jobs had 14.6% (n = 6) frequency of metabolic syndrome, while the subjects with the highest frequency of metabolic syndrome were battery recyclers 38.9% (n = 7), followed by Garbage recyclers (31.6%, n = 17).
The median levels of glucose in control jobs were 79.0 mg/dL (IQR, 60-6-52.0), while the highest levels of glucose were shown in welders (median of 89.0 mg/dL; IQR, 61.0-166). The jobs that showed the highest number of individuals with glycemia greater than 100 mg/dL were welders with 18.2% (n = 22) followed by mechanics with 17% (n = 10).
The presence of basophiles in controls is 24.1%, compared to the highest levels among workers in battery recyclers (46.4%, n = 13) (Table 2).
The analysis shows that 35% (n = 102) of the population have a BMI Ë‚ 25 kg/m2, 45% (n = 131) with overweight (25-30 kg/m2), and 19.9% (n = 58) in obesity condition (≥30 kg/m2). The median body mass index in controls was 26.1 (IQR, 16.7-35.0), with no significant differences between type of informal labor (Table 2).
Controls had higher cholesterol levels (P<0.001), and lower blood glucose levels (P-value < 0.001). The full comparisons are listed in Table 1-2.
In addition, major concerns represent the statistical analyses in the manuscript (especially the logistic regression models) which do not seem appropriate in order to assess the aim of the study, which is the assessment of the health changes in the Pb and Cd exposed workers.
Reviewer #1 - Response #6: We thank the reviewer for the comment. Logistic regression is just a method to assess differences in high levels of Pb and Cd in exposed workers, showing these workers have increased levels of Pb and Cd. Taking into consideration other methods we additionally report the P-value of the chi-square test for detection of differences in Table 1, showing Pb is increased in Mechanics, Battery recyclers and welders.
Although the data obtained offer great potential, it seems that the authors failed to appropriately assess the possible association of health-related parameters (cholesterol, blood pressure, determination of metabolic syndrome, glucose levels, etc.) with actual Pb/Cd exposure in occupational settings and in controls, and as such, to assess the possible effect of Pb and Cd on the health of workers. It appears that the study rather focuses on Pb and Cd exposure assessment in separate types of work (as seen based on Table 4 and abstract).
Reviewer #1 - Response #7: We thank the reviewer for the comment. We did assess the relationship between laboratory parameters and elevated lead levels. These associations are listed in Table 3.
Furthermore, the English language of the manuscript needs to be significantly improved.
Reviewer #1 - Response #8: We thank the reviewer for the comment. We have reviewed the English of the manuscript.
Therefore, at this point, my decision would be to reject the manuscript for publication.
Reviewer 2 Report
The paper “Changes in the health and laboratory parameters in workers with chronical exposure to lead and cadmium” by F. Vergara-Murillo et al. exposes a study about the health and clinical parameters to lead and cadmium levels in chronically exposed workers.
The proposal is good, the study seems well-conducted, but the paper requires extensive revision before it can be revised again to decide if it can be published: there are a lot of writing errors and missing information. The discussion section must be rewritten, including a more detailed explanation of the obtained results.
Abstract:
Line 11: “laboratory parameters” should be changed to “clinical values” or something similar.
Line 17: include units in the 1.22 value for Cd (they should be microg/L and are not the same that for Pb, microg/dL). The confusion with Cd units (sometimes microg/L, sometimes microg/dL) appears over the entire document and creates a big confusion.
In addition, although both symbols are valid, it is preferred “L” over “l” for readability reasons.
1.- Introduction:
Line 30-31. Add a similar short comment about entrance and distribution for Cd.
Lines 32-38. Add a similar comment about sources of Pb.
Line 39: Define what you mean by “informal worker”.
I do not know if there is a better word to define this concept in English (perhaps “illegal employment” or “undeclared work”).
Line 45: you have jumped from reference [11] to reference [20-22], and later on there is no logical sequence in the order of the references. Check it carefully.
2.- Materials and methods.
Line 73-74: what happens about the control workers and medication? Has it been checked?
Line 81: if vacutainer is a registered brand, it must be used © every time it appears. In another case, remove of all them.
Line 95-105 2.3 Chemical and reagents: several mistakes appear in the text. Sub-indexes are missing in all the chemical formulas. Ammonium phosphate is written differently in line 96 and line 106, and in both cases in the wrong way. I suppose that you have used ammonium dihydrogen phosphate, (NH4)(H2PO4). Please check and correct.
In addition, Millipore-grade water does not exist. The water obtained with a Millipore system is Type 1 Water –ASTM D1193- or Grade 1 water -ISO 3696-, or colloquially, ultrapure water.
Lines 118-130 2.5. Quality control and quality assurance. Please, add in supplementary information the obtained results for all the performed tests. For the CRM results, indicate clearly what is the meaning for the percentages values (recovery?) References 5, 24, and 25 do not expose any analytical method, or not in detail. Provide a reference for the employed method.
Lines 131-141. 2.6 Statistical analysis. Kruskal-Wallis test: have you checked, before using it, that the conditions to apply it are fulfilled? (homoscedasticity, same values distribution). Add some additional information about the logistic regression you have employed.
Line 137. I think that there is an error in the high-level values presented. Pb is measured in microg/dL, but Cd in microg/L. But you use the same value, 5 microg/dL for both. Check it. In addition, how have you selected these values? The Biological Limit Values (BLVs) for evaluating potential health risks in occupational health are about 5 microg/L (blood concentration) for cadmium in several countries. However, for lead, this value is much higher: 30-70 microg/dL. You must explain why and from where have you selected these values. The conclusions of your work could change.
Line 144-147: revise the results for BMI. It is impossible a median value of 130 and an IQR of 90-190. I am in shock with an 83 years old worker. What is this man working on?
Line 157-159: please, add a superscript to squared meters, m2.
Line 166: please, add concentration units for Cd values.
Line 169-170: perhaps the relationship between lead and basophils should be explained in the Discussion section.
Line 171-172: the higher levels of Cd in control samples require some discussion in that section.
Table 1. In the “Garbage recycler” column, there is an extra “recycler” in the head file.
Some of the classes you use must be defined: i. e. what is the meaning of “very poor, poor, middle class” in economic status.
At the bottom of the table, perhaps Cd concentration must be microg/L, not g/dL.
Discussion section: you must refer more specifically to your results, and establish connections with previously published papers. There is more information coming from outside than from your data.
In some aspects, it is debatable your conclusions. For example, lines 220-222. Relationship between the lack of education and the chronic exposure to lead. Perhaps the problem is not the lack of education but the lack of formation in their jobs, primarily if they belong to the underground economy. The lack of education probably is related because this factor implies the worst jobs.
You must rewrite this part of the paper, focusing on your data, and comparing them with other relevant studies.
Line 258. Author contribution. Reduce by using initials of the names.
Line 275. Revise the sentence. It is unclear.
Author Response
The paper “Changes in the health and laboratory parameters in workers with chronical exposure to lead and cadmium” by F. Vergara-Murillo et al. exposes a study about the health and clinical parameters to lead and cadmium levels in chronically exposed workers.
The proposal is good, the study seems well-conducted, but the paper requires extensive revision before it can be revised again to decide if it can be published: there are a lot of writing errors and missing information. The discussion section must be rewritten, including a more detailed explanation of the obtained results.
Reviewer #2 - Response #1: We thank the reviewer for the recommendations.
Abstract:
Line 11: “laboratory parameters” should be changed to “clinical values” or something similar.
Reviewer #2 - Response #2: We thank the reviewer for the recommendation. We changed the word.
Line 17: include units in the 1.22 value for Cd (they should be microg/L and are not the same that for Pb, microg/dL). The confusion with Cd units (sometimes microg/L, sometimes microg/dL) appears over the entire document and creates a big confusion.
Reviewer #2 - Response #3: We thank the reviewer for the recommendation. We made a type in the statistical section. We thank the reviewer for spotting this. We change this sentence to:
we defined levels higher than 5µg/dL of lead and 5µg/L cadmium as high
In addition, although both symbols are valid, it is preferred “L” over “l” for readability reasons.
Reviewer #2 - Response #4: We thank the reviewer for the recommendation. We change the L to uppercase.
1.- Introduction:
Line 30-31. Add a similar short comment about entrance and distribution for Cd.
Reviewer #2 - Response #5: We thank the reviewer for the recommendation. We added the following to introduction:
The chronic exposure to heavy metals is an important source of burden of disease worldwide[1]. Lead and cadmium are very important heavy metals that cause chronic diseases for their exposure [2], these metals are among the most frequent heavy metals reported as occupational intoxication worldwide [3–5]. Lead is distributed mainly to blood, soft tissues, and bones after its absorption from lung or gastrointestinal tract [6]. Lead (Pb) has been one of the main pollutants in human health [7]. Routes of occupational and environmental exposure to lead are digestion, absorption, and inhalation[8]. Scientific evidence points out that exposure to lead causes changes in DNA methylation causing alterations in DNA methyltransferases, increased concentration of oxidative stress, alterations in the endocrine system, metabolic syndrome, increased risk of cardiovascular disease, increased blood pressure and increased risk of cancer [7]. A study that relates the risks of exposure to lead with health outcomes, indicates that chronic exposure for 28 days affects hematological parameters and lipid profile in mice [9].
Chronical exposure to lead in occupational settings causes accumulation in soft tissues, bones, and developing toxic effects including neurotoxic[3,6], cognitive dysfunction in older adults [10,11], renal dysfunction[12], cardiovascular disease and hypertension [13], and reproductive organ dysfunction in males and females [14].
The adverse health effects of high blood level of lead and cadmium are well known. Around half of the working population in Latin American countries are in informal workers Cadmium may inhibit protein synthesis in the human body, alternate carbohydrate metabolism, and liver enzymes[1]
Anthropogenic sources of cadmium toxicity, those produce by human activities, come from non-ferrous metal mining and refining, the uses of phosphate fertilizers, the fossil fuel use and the waste incineration or disposal [15,16]. Cadmium have been used in plating, fabrication of nickel-cadmium batteries, anticorrosive agent, stabilizers in plastic materials, allows, and color pigments. The chronical exposure to cadmium has been reported in most world countries [15,17–21].
Environmental exposure to cadmium occurs through food consumption, occupational industries, or smoking [3]. Cd is not an essential element for humans, and Cd exposure has a strong impact in the human health, affecting liver, kidney, and bones. The oxidative stress and tissue damage are mechanisms of cadmium toxicity.
One the most important effect of the Cd exposure is in the osseous system. In rats, osteoporosis has been associated with high levels of cadmium exposure[9]. Cadmium exposure has been associated with hypertension in humans and several animal models. However, the mechanisms by which Cd induces hypertension have yet to be completely elucidated.
Several potential mechanisms include increased oxidative stress, disruption of calcium signaling, kidney damage, interference with the renin–angiotensin system, and dysfunction and impairment of the vascular endothelium[22].
Around half of the working population in Latin American countries are in informal workers [22,23]. These informal jobs can lead to an even greater risk of organs injury from chronical exposure to lead and cadmium [3]. Despite this, only a few reports address this problem in Latin America and the Caribbean. Exposure to lead or cadmium is primarily through inhalation of fumes at work, and accidental ingestion during drinking or eating on site via hands-to-mouth and contacting with contaminated hands as well as wearing dirty clothes [22,24,25]
In low- and middle-income countries, a large percentage of employment is “informal”; these jobs do no declare income to the state, therefore, they do not have the legal protections and guaranties of the government.
Lines 32-38. Add a similar comment about sources of Pb.
Reviewer #2 - Response #6: We thank the reviewer for the recommendation. We added to introduction, as listed un Response #7 of Reviewer 2.
Line 39: Define what you mean by “informal worker”.I do not know if there is a better word to define this concept in English (perhaps “illegal employment” or “undeclared work”).
Reviewer #2 - Response #7: We thank the reviewer for the needed clarification. We have added a paragraph to the introduction, as follows:
In low- and middle-income countries, a large percentage of employment is “informal”; these jobs do no declare income to the state, therefore, they do not have the legal protections and guaranties of the government.
Line 45: you have jumped from reference [11] to reference [20-22], and later on there is no logical sequence in the order of the references. Check it carefully.
Reviewer #2 - Response #8: We thank the reviewer for spotting this. We have carefully reviewed and corrected this.
2.- Materials and methods.
Line 73-74: what happens about the control workers and medication? Has it been checked?
Reviewer #2 - Response #9: We thank the reviewer for this comment. We have reviewed our data, and we chose controls on the bases they did not take any medications. We have modified the text, as follows:
None of the non-control subjects studied were taking any medication.
Line 81: if vacutainer is a registered brand, it must be used © every time it appears. In another case, remove of all them.
Reviewer #2 - Response #10: We thank the reviewer for this clarification. We have added the ©.
Line 95-105 2.3 Chemical and reagents: several mistakes appear in the text. Sub-indexes are missing in all the chemical formulas. Ammonium phosphate is written differently in line 96 and line 106, and in both cases in the wrong way. I suppose that you have used ammonium dihydrogen phosphate, (NH4)(H2PO4). Please check and correct.
Reviewer #2 - Response #11: We thank the reviewer for the detecting the typo, and apologize for this. We have corrected the formula.
In addition, Millipore-grade water does not exist. The water obtained with a Millipore system is Type 1 Water –ASTM D1193- or Grade 1 water -ISO 3696-, or colloquially, ultrapure water.
Reviewer #2 - Response #12: We have reviewed this wording and have corrected this.
Lines 118-130 2.5. Quality control and quality assurance. Please, add in supplementary information the obtained results for all the performed tests. For the CRM results, indicate clearly what is the meaning for the percentages values (recovery?) References 5, 24, and 25 do not expose any analytical method, or not in detail. Provide a reference for the employed method.
Reviewer #2 - Response #13: We thank the reviewer for the recommendation. We have added this to the Supplementary Material.
Lines 131-141. 2.6 Statistical analysis. Kruskal-Wallis test: have you checked, before using it, that the conditions to apply it are fulfilled? (homoscedasticity, same values distribution). Add some additional information about the logistic regression you have employed.
Reviewer #2 - Response #14: We thank the reviewer for the recommendation. We did find some of the tests did not fully fulfill the assumptions of the Kruskal-Wallis test. We have added an indicator in the Tables signaling this no fulfillment of the homoscedasticity assumption.
We also visually inspected the median with box-plots to assess distributional changes in the shape of the distributions of medians, finding all analyses fulfill this assumption.
Line 137. I think that there is an error in the high-level values presented. Pb is measured in microg/dL, but Cd in microg/L. But you use the same value, 5 microg/dL for both. Check it. In addition, how have you selected these values? The Biological Limit Values (BLVs) for evaluating potential health risks in occupational health are about 5 microg/L (blood concentration) for cadmium in several countries. However, for lead, this value is much higher: 30-70 microg/dL. You must explain why and from where have you selected these values. The conclusions of your work could change.
Reviewer #2 - Response #15: We thank the reviewer for the recommendation. We followed the CDC recommendations for chronic exposure to lead (https://www.cdc.gov/nceh/lead/advisory/acclpp/actions-blls.htm). These limits are for surveillance of chronic exposure.
Line 144-147: revise the results for BMI. It is impossible a median value of 130 and an IQR of 90-190. I am in shock with an 83 years old worker. What is this man working on?
Reviewer #2 - Response #16: We thank the reviewer for the recommendation. We have corrected the BMI values. About the age, we made a typo in writing results and reported a patient of 77 as 83 years. This patient was a control. In Table 1, first row, we show the ranges of age in the entire sample. The highest age for the informal works were 68 years old. We found 6% of informal work was made in population greater than 60 years old (14/236).
Line 157-159: please, add a superscript to squared meters, m2.
Reviewer #2 - Response #17: We thank the reviewer for the recommendation. We have added the superscript.
Line 166: please, add concentration units for Cd values.
Reviewer #2 - Response #18: We thank the reviewer for the recommendation. We have added the concentration unit.
Line 169-170: perhaps the relationship between lead and basophils should be explained in the Discussion section.
Reviewer #2 - Response #19: We thank the reviewer for the recommendation. We have added a comment in the discussion about this issues, as follows:
Evidence from in vivo studies associates lead toxicity with the appearance of basophils, the production and increase of free radicals, including reactive oxygen species (ROS), hydroperoxides, singlet oxygen and hydrogen peroxide, depleting glutathione reserves[32]. These facts concur with the results showing the presence of basophils associated with high levels of lead in whole blood.
Line 171-172: the higher levels of Cd in control samples require some discussion in that section.
Reviewer #2 - Response #20: We thank the reviewer for the recommendation. We have added a comment in the discussion about this issues, as follows:
Exposure to cadmium has been related to cigarette consumption, a person who consumes 20 cigarettes a day is exposed to 1µg of Cd / day, most of it is absorbed in the lungs, which is why our statistical analysis of exposure a Cd were adjusted for cigarette consumption [37]. We found increased levels of cadmium in control jobs, suggesting chronic exposure to the informal jobs studied do not appear to increase cadmium levels.
Table 1. In the “Garbage recycler” column, there is an extra “recycler” in the head file.
Reviewer #2 - Response #20: We thank the reviewer for the recommendation. We have deleted the typo.
Some of the classes you use must be defined: i. e. what is the meaning of “very poor, poor, middle class” in economic status.
Reviewer #2 - Response #21: We thank the reviewer for the recommendation. We have added the following to the text to account for this:
In Colombia, all neighborhoods from each city are classified according to their socio-economical stratum from 1 to 6 (1 is very poor, 2 is poor, 3 is middle-class, 4 is up-per-middle class). We took this governmental classification and used them to characterize the income of the population.
At the bottom of the table, perhaps Cd concentration must be microg/L, not g/dL.
Reviewer #2 - Response #22: We thank the reviewer for detecting the typos. We have corrected them.
Discussion section: you must refer more specifically to your results, and establish connections with previously published papers. There is more information coming from outside than from your data.
Reviewer #2 - Response #23: We thank the reviewer for the recommendation. We have expanded the discussion to account for your kind recommendation.
In some aspects, it is debatable your conclusions. For example, lines 220-222. Relationship between the lack of education and the chronic exposure to lead. Perhaps the problem is not the lack of education but the lack of formation in their jobs, primarily if they belong to the underground economy. The lack of education probably is related because this factor implies the worst jobs.
You must rewrite this part of the paper, focusing on your data, and comparing them with other relevant studies.
Reviewer #2 - Response #24: We thank the reviewer for the recommendation. We have expanded the discussion to account for your suggestion.
Line 258. Author contribution. Reduce by using initials of the names.
Reviewer #2 - Response #25: Done.
Line 275. Revise the sentence. It is unclear.
Reviewer #2 - Response #26: Done.
Round 2
Reviewer 1 Report
The manuscript »Changes in the health and laboratory parameters in workers with chronic exposure to lead and cadmium« has improved, however in my opinion the changes to the manuscript are not sufficient for the publications in the journal. The Abstract, Results, and Discussion parts still need substantial improvement. Below I added some comments.
ABSTRACT:
- Lines 17 – 19: please add what these high levels are! For somebody who will read the abstract, this sentence tells nothing without reading the whole manuscript. Rather write that 29,5% of individuals exceeded the 5μg/dL value, which you set as the limit for high exposure.
- What about cadmium – if levels were low or not elevated in workers, that is still a result and should be summarized here.
- Line 17: delete dot before the median levels of blood cadmium; moreover, the units for Cd are missing here.
- Again, the title of the manuscript is changes in the health and laboratory parameters, however, no information on the possible influence of Cd or Pb exposure on health or laboratory parameters in workers is given in the abstract – in my opinion that should be crucial information of the abstract.
INTRODUCTION:
- The Introduction still needs to be improved: the order of information’s should be corrected : First, give information on anthropogenic and environmental sources – so how the human population is exposed, then what are the routes of exposure, then accumulation in the human body and then give the possible effects on human health. Moreover, there is much more focus on Cd than on Pb in the introduction – should be more balanced as they are equally important for the manuscript.
- Some information now given separately for Cd and Pb could be joined – an example is exposure routs: lines 32-33 and 53-54 – both are true for Pb and for Cd.
- I suggest changing the term heavy metals (all over manuscript) with potentially toxic metals or elements! The term heavy metals was by several publications pointed out as not suitable and misleading. (examples: doi: 3390/ijerph16224446; doi:10.1080/02772248.2017.1413652; https://www.imwa.info/docs/Heavy_Metals_A_meaningless_term.pdf)
- Lines 28 – 30: Please change sentences as: “Lead (Pb) and cadmium (Cd) are among the most frequently reported occupational toxicants worldwide and are consequently associated with various chronic diseases.“
- Line 32: Lead (Pb) has been one of the main pollutants in human health – this sentence is repetition from above – please delete;
- Instead of lead and cadmium use the abbreviations of Pb and Cd throughout the manuscript.
- Line 40-43: Point out that Cd is mainly accumulated in kidneys (and liver). Among toxic effects, renal dysfunction should be mentioned first as kidneys are the primary target of Cd toxicity. Moreover, accumulation is not only in chronical occupational but environmental exposure as well – the same as for Pb.
MATERIAL and METHODS:
- Line 91: I still think you should add a number of participants – it should be stated already here not only in the results
- Line 110: for controls add a number of participants and age, and gender (as you wrote in response to my reviews they were all women).
- Line 121-128 is duplicated in paragraph 2.4; please delete 121-128 lines.
- Line 190: typo (gioven – given)
- Line 198: based on what did you choose those values – add references and explain more in detail why those levels.
RESULTS:
- 1 Population characteristics: please give information on a number of participants, age range and median, smoking, etc. also for the control group! And then report what was significantly different between both groups. Moreover, I would here expect that you report on differences in basic characteristics also between the types of work. Sex parameter is missing in Table 1 – to my previous review, you responded controls were all women –unless this is a mistake - you should specify this in table 1.
- 2 section: Table 2 should be introduced at the beginning already; In table 2 add what is the normal (reference) range for all those parameters, so that the reader can see whether workers have alarmed values for those parameters or they are still in the normal range.
- Table 1: typo in PB – correct to Pb; also write cadmium with Cd – make this consistent throughout the whole article.
- Table 3:
-first column n(%) however, you report only one number – no percentages - correct.
-how were those parameters selected (why is there no glucose included as it was significantly different in table 1; as well as other parameters)
-In the case of Basophile, (%) the data you present is confusing - what are you presenting here median and range?
-* you say as otherwise specified- but you did not specify what this is – should be specified in the table!
-Also * is twice in the footnotes of the table – correct.
-explain also HDL in footnotes
- Lines 235 – 236: you report here Cd and Pb in the same units; however, they are not as such in table 1.
- Line 237: High lead levels were 29.5% - the same as commented for abstract – please write here about % of individuals that exceeded the level of 5 μg/dL set as high level.
- Figure 1: why is this figure not cited and commented in section 3.2 – where laboratory parameters are reported and presented; Moreover, Figures 1 and 2 should be reversed!
- Line 241 – 243: But the levels in workers were in general significantly higher than in control (Table 1 and Figure 2)! Moreover, this should be compared with the exclusion of smokers at this might bias the comparison. It might be that there is more smokers in control group.
- It is still not clear why did you not assess the influence of Pb and Cd on laboratory parameters with the adjustment for smoking etc...the same as you did in Table 4. This should be the main aim of the manuscript and is also expected based on the title of the manuscript. Either you change the title o manuscript or you assess the relationship of Pb/Cd exposure and health-related parameters more in detail. Did the elevation in the Pb or Cd levels influence for example platelets or basophils even if you adjust for smoking, age, years of working?
DISCUSSION:
- The discussion part needs major improvement - there is no clear developed discussion. Example: discussion on Pb and basophils, then importance of education on Pb exposure, followed by the importance of monitoring Pb and Cd in informal workers of Latin America, and then Cd exposure and smoking, and then again platelets and lead.
- Line 275: specify based on which laboratory parameters you saw increased risk of adverse health
- you should discuss what are the normal ranges of platelet and basophiles in human populations and whether those observed in your population are alarming - possibly leading to adverse health effects and which ones and support it with literature.
- Line 287 – 289: this is a very strong statement and not supported by your results. Especially since you did not report any significant changes in laboratory parameters based on Cd levels. How do your results indicate that high Cd and Pb are especially harmful to the kidney and liver?
- Line 289 – 290: this is a repetition (285 – 286).
- Line 314 – 315: This is not supported by your results - levels of Cd are higher in workers (look at table 1 medians).
Reviewer 2 Report
I acknowledge the authors that this revised version has been improved a lot with respect the the first submission. However, there are still some details and minor remarks to correct.
Statistical analysis, lines 200-201. You must explain better how do you adjusted the regression by confounding variables, and why you have selected these variables as confounding.
I still have problems with the limit of 5 microg/dL for Pb. In your answer, you cite CDC recommendations (but you do not include this reference in bibliography). However, if you check carefully the title of that document, it is referred to children. Once again, the occupational value for workers is usually larger, and this must be clearly detailed in the text, because it affects your conclusions.
In addition, please correct:
Abstract: line 17, Cd units are still missing.
Chemical and reagents: recheck some subscripts that appear as superscripts or in plain form (lines 134, 135, 140, 145).
Instrumental analysis: line 162, cite that the methodology is explained in Supplementary material (and where).
Quality control…: again, cite that the results are presented in Supplementary material (and where). Lines 182-184: the percentage is confuse, if the reader does not go to table S2. Please indicate the meaning for these percentages in the text.
Line 236: microg/dL for Cd must be microg/L.
Table 1, parameter Pb, microg/dL. Please, correct that Pb now appears as PB.
Table 1, the remark about BMI must be deleted.
Tables 1-3: please, do not use an asterisk for different meanings. Repeat, if necessary, for different tables the same things (for example, the meaning of one or two asterisk in significance level)